# Evaluation of Clinical Results regarding Peristomal Skin Health Associated with the Adjustment and Formulation of the New Moderma Flex One-Piece Ostomy Devices

**DOI:** 10.3390/jpm13020219

**Published:** 2023-01-26

**Authors:** Sebastian Rivera García, Esperanza-Macarena Espejo Lunar, Julian Rodríguez-Almagro, Silvia Louzao Méndez

**Affiliations:** 1Stomaterapeuta Nurse, University Hospital Virgen del Rocío de Sevilla, 41013 Seville, Spain; 2Faculty of Nursing, University of Castilla-La Mancha, 13001 Ciudad Real, Spain; 3University Hospital of A Coruña, 15006 A Coruña, Spain

**Keywords:** ostomy, nurse, flex moderma, peristomal skin

## Abstract

In order to determine the perception of ostomized patients about the performance and safety of the new one-piece device Moderma Flex, as well as the evolution of peristomal skin health after its use. The pre- and post-experimental multicenter study after the use of the Moderma Flex one-piece ostomy device on 306 ostomized people from 68 hospitals in Spain. We used a self-made questionnaire on the usefulness of different parts of the device and the perception of peristomal skin improvement. The sample was composed of 54.6% (167) men and had an average age of 64.5 years (standard deviation = 15.43). The type of device most commonly used according to its opening was closed by 45.1% (138). In addition, for the type of barrier, the most frequently used is the flat one; 47.7% (146) and 38.9% (119) used a model of soft convexity. A total of 48% scored with the highest assessment in the perception of skin improvement. The percentage of patients with peristomal skin problems decreased from 35.9% at the first visit to less than 8% after the use of Moderma Flex. Further, 92.4% (257) had no skin problems, the most frequent being erythema. The use of the Moderma Flex device seems to be related to a reduction in peristomal skin complications and a perception of improvement.

## 1. Introduction

The performance of an ostomy consists of a surgical procedure in which a hollow organ is connected to the outside. The creation of an ostomy may be necessary temporarily or permanently for the treatment of certain diseases of the digestive and urinary systems [1]. When a person has an ostomy, they suffer a series of biological, functional, psychological, and social alterations that make it difficult for them to adapt to their environment and impact their quality of life [2,3,4]. In addition, with this alteration in the elimination of feces/urine, they have to face hygienic and dietary changes and the loss of their control over sphincters, which can affect their perception of their own image. Further, they may even have difficulties reintegrating into their work, sexual, and social lives [5,6,7,8]. The performance of an ostomy implies the need for the person to acquire new skills for self-care of his stoma, such as correctly removing the pouch without causing injuries, performing proper hygiene on the stoma, maintaining the integrity and good health of the peristomal skin, properly placing the suitable pouch, or disposing of the pouch’s residue. For this reason, the selection of the ostomy device to use according to the needs of each person is fundamental to its reincorporation into their daily lives. The ostomy devices should be safe and easy to handle, acting as a barrier and preventing direct contact of stool or urine with the skin, while being comfortable and discreet so they do not alter the lifestyle of the ostomized person. On the other hand, there are multiple potential complications that can result from the placement of an ostomy, with skin problems being the most frequent. These can occur in between 3.1% and 46% of patients, depending on the location of the stoma and the study consulted [9,10,11,12].

In this context, several studies have reported that the safety against leaks and the protection and care of peristomal skin are two fundamental factors of great concern for the ostomized patient [13,14]. It is especially important to make the correct adjustment of the barrier around the stoma for a proper seal. The solutions, such as soft convexity or the adhesive border built into the hydrocolloid barrier, are designed to provide better adjustment and fit around the stoma. In this way, the correct adaptation to the peristomal skin and the different abdominal contours could potentially prevent leakage and, consequently, the contact of aggressive effluents with the skin.

As a result of all these reasons, this study aims to determine the perception of ostomized patients about the performance and safety of the new one-piece device Moderma Flex^®^ (Hollister, Madrid, Spain), widely used in other countries such as France and the United Kingdom [15,16], but recently incorporated in Spain, as well as the evolution of peristomal skin health after use.

## 2. Material and Methods

### 2.1. Design and Participants

This is a four-dimensional, pre-post, multicenter study on the use of the one-piece Moderma Flex ostomy device. This study was approved by the Ethics Committee in Clinical Research (CEIC) of the Virgen Macarena-Virgen del Rocio University Hospitals in Seville (Protocol Number MF-2021-03). We included elderly ostomized patients whose ostomy had been performed in a maximum of the last 12 months. Those patients who did not have sufficient cognitive capacity to participate in the study were excluded, as were patients with a colostomy, ileostomy, or urostomy whose temporality was less than six months, patients who required the use of firm or deep convexity, and those who did not agree to participate in the study. For the estimation of the sample size, the following parameters were considered for studies with paired measures: accepting an alpha risk of 0.05 and a beta risk of 0.2 in a bilateral contrast. A minimum of 251 subjects are required, assuming that the initial proportion of events is 14% (patients with peristomal cutaneous alterations prior to intervention [11]) and at the end, 7% (percentage of patients with peristomal cutaneous alterations after intervention). A 10 percent follow-up loss rate has been estimated.

### 2.2. Procedure for the Collection of Information

The study was carried out with the collaboration of nurses performing ostomy consultations in 68 centers of the network of Spanish hospitals. Each center recruited and followed a minimum of four patients. This recruitment was carried out during the hospital stay of newly operated patients and in outpatient consultations with patients with an existing ostomy. The nurses delivered the fact sheets to the patients, explained the study verbally, resolved doubts, and invited them to participate in this research.

They then completed a questionnaire on basic and clinical socio-demographic data. From then on, the researchers provided the Moderma Flex device (Hollister, Madrid, Spain) free of charge to patients who agreed to participate in the study. After 15 days of follow-up, a second consultation was held to obtain information on the clinical progress of the patient.

### 2.3. Information Sources and Study Variables

In the collection of the information, two questionnaires were used. The first questionnaire contained 13 questions, and the second questionnaire contained 9 questions. Questions 7, 8, and 9 were Likert-type (degree of agreement and utility valued from 1 to 5 points), where statements 11, 6, and 4 were grouped on the barrier, the urostomy device, and the display window, respectively. The first questionnaire collected information on sociodemographic and clinical variables such as the type of ostomy, morphology of the stoma, type of abdomen, temporality of the ostomy, skin type, peristomal skin problems, characteristics of the device used, frequency of change of device, and reason for change of the usual device. The second questionnaire collected information on the type of device used, accessories used, frequency of change, skin reaction to the adhesive edge, peristomal skin condition after use of the new device, functioning and performance of the skin barrier, urostomy device features, display window utility, and device safety by reporting adverse events. The questionnaire is available in the Appendix A.

### 2.4. Statistical Analysis Used

First, a descriptive analysis was performed using absolute and relative frequencies for qualitative variables and the mean with standard deviation for quantitative variables. Then, a bivariate analysis was performed on the improvement of peristomal skin based on device characteristics and time as ostomized, using the Student-Fisher *t* test and Variance Analysis (ANOVA), depending on the number of categories. A *p* value of 0.05 was considered significant. All analyses were performed with the SPSS v.24.0 statistical package.

## 3. Results

The study involved 306 patients in the first part and 294 in the second, after the use of the new Moderma Flex devices. Regarding the most important sociodemographic characteristics of the basic sample, 54.6% (167) were men and had an average age of 64.5 years (standard deviation = 15.43), with a minimum age of 31 and a maximum age of 92 years. As for the time since the ostomy was performed, the group that presented a higher frequency was those who had been between 1 month and 3 months, with 26.5% (81). The most frequent type of ostomy was a colostomy, with 57.3% (175), being permanent in 55.9% (171) and terminal in 69.9% (214). Among the characteristics of the abdomen of the patients, 52.6% presented a round morphology, and approximately 40–50% had a flaccid, globulose, or smooth abdomen. In terms of peristomal skin characteristics, only 4.6% (14) had an oily appearance, damaged 7.2% (22), and irritated 27.8% (85). Among the most frequently observed complications, irritative contact dermatitis occurred in 67 patients (21.9%), followed by mechanical dermatitis in 21 patients (6.9%), and granulomas in 10 patients (3.3%). Finally, only 9.2% presented a condition that posed a risk for peristomal skin, noting that 8 of the 28 patients with the condition were in treatment with chemotherapy (Table 1).

### 3.1. Description of the Device Used in the Initial Visit

The type of device most commonly used according to its opening is a closed 36.3% (111) and 51.0% (156) 2-device. Most used a flat barrier, 52.6% (161) of the cases. As for the accessories used, the belt stood out with 22.9% (70), followed by the hydrocolloid paste with 18.3% (56). On the other hand, they were also asked about the number of times they changed the device, to which they replied that 19.0% (58) changed it more than once a day and 20.9% every day (64). About the reason for change, 24.2% (74) reported leakage, and 9.8% reported irritation (30) (Table 2).

### 3.2. Description of the Moderma Flex Device Used

In the second part of the study, patients began using the one-piece Moderma Flex device with different options. The type of device most commonly used according to its opening was closed by 45.1% (138). As for the type of barrier, although the most frequently used is flat in 47.7% (146) of the cases, 38.9% (119) used a model of soft convexity. As complementary features, 40.1% (118) had an adhesive edge, and 55.4% (118) had a display window. As for the accessories used, the belt stood out with 25.2% (70), followed by hydrocolloid powder and barrier spray with 19.0% (56) and a hydrocolloid ring with 15.6% (46).

On the other hand, they were also asked about the number of times they changed the device. 40.1% (118) changed it more than once a day and 41.2% every day (121), while 16.7% (49) changed it every two days (Table 3).

### 3.3. Evaluation of Improvement, Adverse Events, Evaluation of the Barrier, Viewing Window, and Urostomy after the Use of the New Device

Among the key questions in the evaluation of the new device, they were asked about the degree of improvement of the peristomal skin, and 48% scored with the highest assessment (a lot), while only 6% scored with the most negative assessment (nothing). Regarding the complications observed, the low incidence of these is remarkable. Only 0.3% presented allergies, 2.4% irritation, and 2.7% itching. They were then asked about the function and performance of the barrier. In this sense, all items had an average agreement degree above 4.30 (the range of possible values was between 1 and 5), with the characteristics best valued being “the comfort of the barrier” and “the adaptation to the abdominal contour” with an average of 4.59, followed by “barrier flexibility” with an average of 4.48 (Table 4 and Figure 1).

Additionally, regarding the display window, almost all items had average scores above 4.40, highlighting “the possibility of inspecting the state of the stoma”, which stood at 4.72 on average (Figure 2). The group of urostomized patients was then asked about the functioning and performance of their device. The average scores were above 4.25 in all items (Table 4 and Figure 3). The presence of adverse skin events and their resolution were also evaluated. In this sense, 92.4% (257) presented no skin problem, and a total of 28 events were detected, the most frequent being erythema. That is, the percentage of patients with peristomal skin problems decreased from 35.9% in the first visit to a percentage lower than 8% after the use of Moderma Flex. After the end of the study, only 8 of the patients maintained the detected problem, which had not yet been solved (Table 5).

### 3.4. Subanalysis of Peristomal Skin Enhancement Depending on Device Characteristics and Ostomized Time

Finally, the perception of peristomal skin improvement was analyzed for all patients and for the subgroup of patients with damaged or irritated skin depending on the characteristics of the device. In this sense, only a statistically significant association was observed with the use of an adhesive border, both in all patients (*p* = 0.008) and in patients with damaged or irritated skin (*p* = 0.011). Specifically, those who do not use adhesive border presented higher average improvement scores than those who use it, although in both groups an improvement in the perception of peristomal skin was observed. No statistically significant differences were found in the rest of the comparisons (Table 6 and Table 7).

## 4. Discussion

This multicenter study was carried out in order to know the perception in the improvement of peristomal skin after the use of a new device for one-piece ostomies (Moderma Flex). The patients reported a positive assessment of the improvement of periostomal skin both in those who had it previously intact and without lesions, and in those who had it damaged, coinciding with the results of another previous study [17]. In the same vein, patients’ assessments of the characteristics of the barrier and the viewing window, as well as of the urostomy device, were very positive. No statistical differences were found with the main comparison variables, except for the use of the adhesive border. Specifically, the perception of improved peristomal skin was greater in those who did not use it. This result is perfectly logical and is explained because those patients who require the use of a pouch with an adhesive border are those who present leaks, making the a peristomal skin more damaged [18] and with greater difficulty for its improvement. However, both groups showed an improvement in the perception of peristomal skin. Moreover, the fact that there are no differences in perception according to the rest of the characteristics of the device (type of barrier, type of opening, window) could be attributed to the fact that the devices used have been selected according to the individual needs of each ostomate.

In addition, the percentage of patients with peristomal skin problems has fallen considerably from 35.9% in the first visit to less than 8% after the use of Moderma Flex. These results are very positive and are among those described by other studies. Specifically, Malik et al. [11] in a systematic review of clinical trials found an incidence of skin problems of 14.0% with a confidence interval between 2.4% and 46.2%, higher than the findings of this study. The main limitations of the study include the quasi-experimental design where there is no parallel control group, so some of the improvement could be attributed to the Hawthorne effect [19] and a short follow-up of patients, although the perception of improvement is likely to increase over time. In addition, it would have been advisable to have blinded the device used, although from a practical point of view, it would have been very complex since it is the patients themselves who perform the care of their ostomy and change of device. Otra limitación importante es que los pacientes son muy heterogeneous en cuanto a características clínicas, por lo que no se puede atribuir toda la mejora exclusivamente al empleo del dispositivo. In terms of strengths, the participation in the study of 68 centers stands out, presenting adequate external validity to be included in this sample of people from different geographical areas, health services, and adequate representation of ostomized patients.

Therefore, we can conclude that the use of the Moderma Flex device seems to be related to a reduction of peristomal skin complications and a perception of improvement. However, a longer-term follow-up would be advisable to determine the evolution of peristomal skin health, the quality of life of patients, and the incidence of other adverse events.

## Figures and Tables

**Figure 1 jpm-13-00219-f001:**
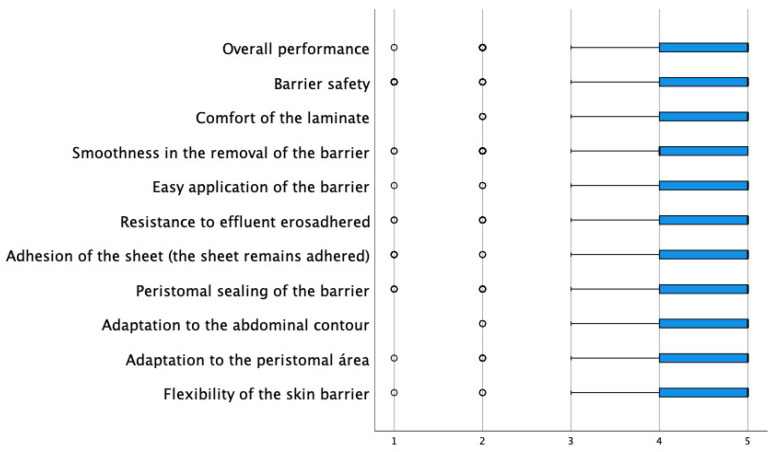
Distribution of scores in the evaluation of the usefulness of the barrier.

**Figure 2 jpm-13-00219-f002:**
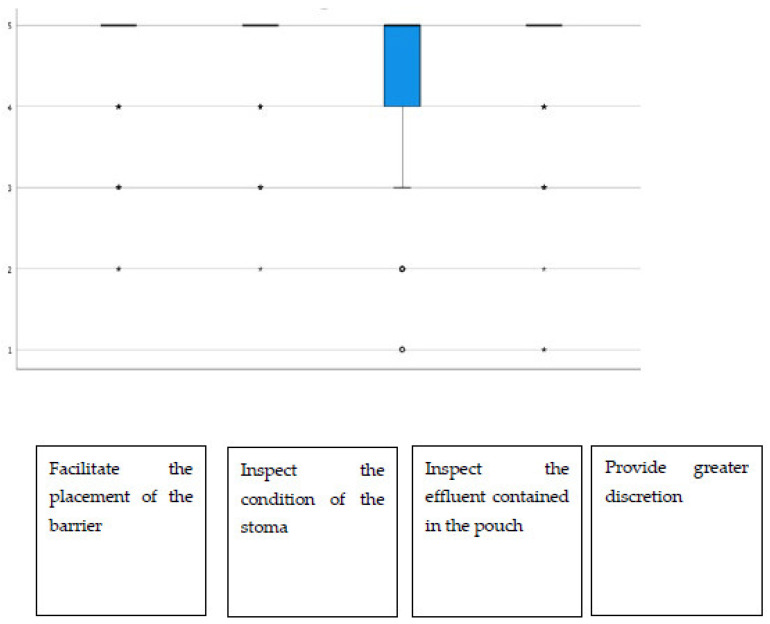
Distribution of scores in the utility rating of the display window.

**Figure 3 jpm-13-00219-f003:**
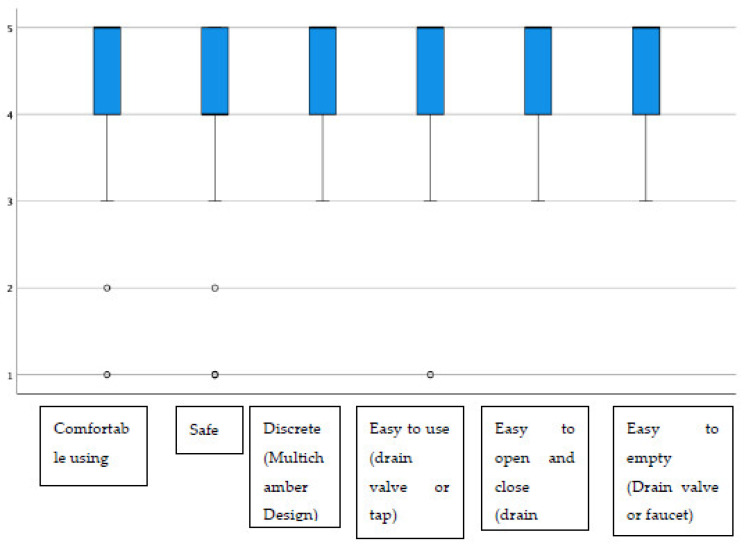
Distribution of scores in evaluating the usefulness of the urostomy device.

**Table 1 jpm-13-00219-t001:** Clinical and sociodemographic characteristics of study subjects.

Clinical and Sociodemographic Characteristics	% (n)
Gender	
Man	54.6 (167)
Woman	45.4 (139)
Age (Years)	64.5 (12.43)
Time from completion of ostomy	
<1 week	8.8 (27)
1 week–1 month	22.5 (69)
1 month–3 months	26.5 (81)
3 months–6 months	19.3 (59)
6 months–12 months	22.9 (70)
Type of ostomy (depending on surgery)	
Urostomy	17.6 (54)
Ileostmy	25.2 (77)
Colostomy	57.2 (175)
Type of ostomy according to duration	
Temporary	44.1 (135)
Permanent	55.9 (171)
Type of ostomy according to technique	
Loop ostomy	20.6 (63)
Termina	69.9 (214)
Other Type	9.5 (29)
Morphology of the stoma	
Round	52.6 (161)
Protruding	32.0 (98)
Oval	27.1 (83)
Flat	29.1 (89)
Irregular	6.9 (21)
Invaginate	8.2 (25)
Edematous	0.7 (2)
Stenosed	1.3 (4)
Type of abdomen	
Smooth	48.4 (148)
Globulous	42.2 (129)
Folds	32.0 (98)
Flaccid	40.8 (125)
Scars	26.1 (80)
Skin Type	
Dry	40.5 (124)
Mixed	54.9 (168)
Oily	4.6 (14)
Skin type according to integrity	
Intact	65.0 (199)
Irritated	27.8 (85)
Damaged	7.2 (22)
Complications observed in the skin	
None	64.1 (196)
Irritative contact dermatitis	21.9 (67)
Mechanical dermatitis	6.9 (21)
Granuloma	3.3 (10)
Pressure ulcer	0.3 (1)
Dehiscence mucocutaneous suture	2.3 (7)
Dermatitis Incrustation	0.3 (1)
Incrustation	0.3 (1)
Neoplasia	0.3 (1)
You have a condition that can put peristomal health at risk	
No	90.8 (278)
Yes	9.2 (28)
Oncological treatment	8
Crohn desease	2
Paraestomal Hernia	2
Ulcerative colitis	1
Atopy/eczema/psoriasis	3

**Table 2 jpm-13-00219-t002:** Characteristics of the device used in the initial visit.

Characteristics of the Device Used	% (n)
Type of device you currently use (opening)	
Closed	36.3 (111)
Drainable	25.2 (77)
With valve	14.7 (45)
Does not apply	23.8 (73)
Device currently in use (by pieces)	
1 piece	25.2 (77)
2 pieces	51.0 (156)
3 pieces	1.6 (5)
Others	0.7 (2)
Does not apply	21.6 (66)
Device currently used according to foil type	
Flat barrier	52.6 (161)
Convex barrier	23.2 (71)
Does not apply	24.2 (74)
Accessories used with the device	
Hydrocolloid rings	17.0 (52)
Hydrocolloid powders	17.6 (54)
Protective strips	2.0 (6)
Hydrocolloid paste	18.3 (56)
Barrier cream	0.0 (0)
Convex ring	3.6 (11)
Belt	22.9 (70)
Skin barrier spray	17.0 (52)
Adhesive remover spray	12.4 (38)
Other accessories	17.0 (52)
Does not apply	26.1 (80)
Number of accessories used	
None	42.8 (131)
One	22.2 (68)
Two	20.9 (64)
Three	11.1 (34)
Four	1.6 (5)
Five	1.0 (3)
Six	0.3 (1)
Frequency of the device change	
More than once a day	19.0 (58)
Every day	20.9 (64)
Every two days	21.9 (67)
Every three days	16.0 (49)
Every four days	2.6 (8)
Does not apply	19.6 (60)
Reason for the device change	
Rutine	50.3 (154)
Leaks	24.2 (74)
Irritation	9.8 (30)
Itching	5.2 (16)
Peeling Sheet	9.2 (28)
Erosion Sheet	2.0 (6)

**Table 3 jpm-13-00219-t003:** Characteristics of the Moderma Flex device evaluated.

Characteristics of the Device Used	% (n)
Type of device you currently use (opening)	
Closed	45.1 (138)
Drainable	34.0 (104)
With valve	17.0 (52)
Missing	12
Device currently used (per sheet)	
Flat barrier	47.7 (146)
Soft convex barrier	38.9 (119)
Moderate convex barrier	9.5 (29)
Missing	12
Adhesive border	
No	59.9 (176)
Yes	40.1 (118)
Missing	12
Viewing window	
No	44.6 (131)
Yes	55.4 (163)
Missing	12
Accessories used with the device	
Hydrocolloid rings	15.6 (46)
Hydrocolloid powders	19.0 (56)
Protective strips	1.7 (5)
Hydrocolloid paste	11.2 (33)
Barrier cream	1.0 (3)
Convex rings	4.1 (12)
Belt	25.2 (74)
Skin barrier spray	19.0 (56)
Adhesive remover spray	17.0 (50)
Other accessories	37.4 (110)
Number of accessories used	
None	38.1 (112)
One	28.6 (84)
Two	20.4 (60)
Three	8.2 (24)
Four	4.1 (12)
Five	0.3 (1)
Six	0.3 (1)
Frequency of the device change	
More than once a day	40.1 (118)
Every day	41.2 (121)
Every two days	16.7 (49)
Every three days	1.7 (5)
Every four days	0.3 (1)
Missing	12

**Table 4 jpm-13-00219-t004:** Evaluation of the improvement, adverse events, and foil and display window after the use of the device.

	Degree of Agreement (Scale 1 Nothing According to 5 Very Agree)					
	Nothing 1 n (%)	Little 2 n (%)	Indifferent 3 n (%)	Fairly 4 n (%)	Very 5 n (%)	Average (DE)
Improvement of the periostomal skin						
All (n = 294)	18 (6.1)	4 (1.4)	43 (14.6)	88 (29.9)	141 (48.8)	4.12 (1.11)
With intact skin (n = 190)	15 (7.9)	2 (1.1)	28 (14.7)	55 (28.9)	90 (47.4)	4.07 (1.17)
With damaged or irritated skin (n = 104)	3 (2.9)	2 (1.9)	15 (14.4)	33 (31.7)	51 (49.0)	4.22 (0.97)
Cutaneous lamina (n = 294)						
Flexibility of the skin barrier	1 (0.3)	2 (0.7)	17 (5.8)	109 (37.1)	165 (56.1)	4.48 (0.67)
Adaptation to the peristomal area	1 (0.3)	3 (1.0)	16 (5.4)	94 (32.0)	180 (61.2)	4.45 (0.68)
Adaptation to the abdominal contour	0 (0.0)	2 (0.7)	15 (4.9)	84 (27.5)	191 (62.4)	4.59 (0.62)
Peristomal sealing of the barrier	3 (1.0)	4 (1.4)	26 (8.5)	103 (35.3)	156 (53.4)	4.39 (0.79)
Adhesion of the blade	4 (1.4)	2 (0.7)	15 (5.1)	89 (30.3)	184 (62.6)	4.52 (0.75)
Resistance to effluent erosion	4 (1.4)	4 (1.4)	27 (9.2)	107 (36.4)	154 (52.4)	4.38 (0.77)
Easy application of the barrier	1 (0.3)	2 (0.7)	19 (6.5)	88 (30.1)	182 (62.3)	4.53 (0.68)
Smoothness in the removal of the barrier	2 (0.7)	8 (2.7)	31 (10.6)	109 (37.2)	143 (48.8)	4.31 (0.82)
Blade comfort	0 (0.0)	2 (1.0)	11 (3.7)	91 (31.0)	189 (64.3)	4.59 (0.62)
Barrier safety	6 (2.0)	4 (1.4)	21 (7.1)	89 (30.3)	174 (59.2)	4.43 (0.85)
Overall performance of the sheet	1 (0.3)	10 (3.4)	12 (4.1)	106 (34.6)	164 (56.0)	4.44 (0.76)
Viewing option (n = 186)						
Facilitate the placement of the barrier	0 (0.0)	3 (1.6)	10 (5.4)	32 (17.2)	141 (75.8)	4.67 (0.65)
Inspect the condition of the stoma	0 (0.0)	1 (0.5)	9 (4.8)	32 (17.2)	146 (77.7)	4.72 (0.58)
Inspect the effluent contained in the pouch	2 (1.1)	6 (3.2)	15 (8.0)	45 (24.1)	119 (63.6)	4.46 (0.86)
Provide greater discretion	3 (1.6)	1 (0.5)	11 (5.9)	29 (15.5)	143 (76.5)	4.65 (0.76)
Facilitate the placement of the barrier	0 (0.0)	3 (1.6)	10 (5.4)	32 (17.2)	141 (75.8)	4.67 (0.65)
Inspect the condition of the stoma	0 (0.0)	1 (0.5)	9 (4.8)	32 (17.2)	146 (77.7)	4.72 (0.58)
Urostomy device (N = 58)						
Comfortable using	1 (1.7)	1 (1.7)	4 (6.9)	18 (31.0)	34 (58.6)	4.43 (0.84)
Safe	3 (5.2)	1 (1.7)	2 (3.4)	24 (41.4)	28 (48.3)	4.26 (1.00)
Discrete (multichamber design)	0 (0.0)	0 (0.0)	10 (17.5)	13 (22.8)	34 (59.6)	4.42 (0.78)
Easy to use (drain valve or tap)	1 (1.8)	0 (0.0)	3 (5.3)	21 (36.8)	32 (56.1)	4.46 (0.76)
Easy to open and close (drain valve or tap)	0 (0.0)	0 (0.0)	2 (3.6)	22 (39.3)	32 (57.1)	4.54 (0.57)
Easy to empty (drain valve or tap)	0 (0.0)	0 (0.0)	1 (1.8)	19 (33.3)	37 (64.9)	4.63 (0.52)

**Table 5 jpm-13-00219-t005:** Adverse events and resolution.

Adverse Event	Not Recovered Yet% (n)	Recovered % (n)	Total N = 278 % (n)
None			92.4 (257)
Erythema	12.5 (1)	87.5 (7)	2.7 (8)
Erosion	100 (2)	0.0 (0)	0.7 (2)
Hypergranulation	100 (2)	0.0 (0)	0.7 (2)
Necrotic tissue	50.0 (1)	50.0 (1)	0.7 (2)
Leakage	33.3 (1)	66.6 (2)	1.02 (3)
Prolapse	100 (1)	0.0 (0)	0.34 (1)
Rod	0.0 (0)	100 (1)	0.34 (1)
Irritation	0.0 (0)	100 (1)	0.34 (1)
Pruritus	0.0 (0)	100 (1)	0.34 (1)

**Table 6 jpm-13-00219-t006:** Evaluation of peristomal skin improvement in all patients according to the type of device and some characteristics.

	Has Peristomal Skin Improved? (Scale 1 Nothing According to 5 Very Agree) (N = 294)						Value *p*
	Nothing 1 n (%)	Little 2 n (%)	Indifferent 3 n (%)	Fairly 4 n (%)	Very 5 n (%)	Average (DE)	
Type of device you currently use (opening)							0.232
Close	10 (7.2)	1 (0.7)	15 (10.9)	32 (23.2)	80 (58.0)	4.24 (1.15)	
Open	5 (4.8)	1 81.0)	20 (19.2)	40 (38.5)	38 (36.5)	4.01 (1.02)	
Drain valve	3 (5.8)	2 (3.8)	8 (15.4)	16 (30.8)	23 (44.2)	4.04 (1.14)	
Device currently used according to the type of sheet							0.244
Flat	7 (4.8)	1 (0.7)	23 (15.8)	39 (26.7)	76 (52.1)	4.21 (1.05)	
Soft Convexity	10 (8.4)	3 (2.5)	16 (13.4)	39 (32.8)	51 (42.9)	3.99 (1.20)	
Moderate Convexity	1 (3.4)	0 (0.0)	4 (13.8)	10 (34.5)	14 (48.3)	4.24 (0.95)	
Adhesive Border							0.008
No	10 (5.7)	2 (1.1)	21 (11.9)	42 (23.9)	101 (57.4)	4.26 (1.09)	
Yes	8 (6.8)	1 (1.7)	22 (18.6)	46 (39.0)	40 (33.9)	3.92 (1.09)	
Viewing option							0.593
No	7 (5.3)	2 (1.5)	22 (16.8)	42 (32.1)	58 (44.3)	4.08 (1.07)	
Yes	11 (6.7)	2 (1.2)	21 (12.9)	46 (28.2)	83 (50.9)	4.15 (1.31)	
Ostomized time							0.236
<1 week	2 (7.4)	0 (0.0)	0 (0.0)	7 (25.9)	18 (66.7)	4.44 (1.09)	
1 week–1 month	4 (5.9)	1 (1.5)	14 (20.6)	27 (39.7)	22 (32.4)	3.91 (1.06)	
1 month–3 months	4 (5.9)	1 (1.3)	11 (14.3)	24 (31.2)	37 (48.1)	4.16 (1.07)	
3 months–6 months	3 (5.5)	1 (1.8)	6 (10.9)	15 (27.3)	30 (54.5)	4.24 (1.09)	
6 months–12 months	5 (7.5)	1 (1.5)	12 (17.9)	15 (22.4)	34 (50.7)	4.07 (1.20)	

**Table 7 jpm-13-00219-t007:** Evaluation of the improvement of peristomal skin in patients with damaged and irritated skin according to the type of device and some characteristics.

	Has the Peristomal Skin Improved? (Scale 1 Nothing According to 5 Very Agree) (N = 104)						Value *p*
	Nothing 1 n (%)	Little 2 n (%)	Indifferent 3 n (%)	Fairly 4 n (%)	Very 5 n (%)	Average (DE)	
Type of device you currently use (opening)							0.230
Closed	1 (2.8)	0 (0.0)	2 (5.6)	1 (25.6)	23 (53.5)	4.44 (0.84)	
Drainable	1 (2.0)	1 (2.0)	8 (15.7)	18 (35.3)	23 (45.2)	4.11 (0.92)	
With valve	0 (0.0)	0 (0.0)	1 (10.0)	4 (40.0)	5 (50.0)	4.09 (1.19)	
Device currently used (per sheet)							0.828
Flat	2 (4.7)	1 (2.3)	6 (14.0)	11 (25.6)	23 (53.5)	4.21 (1.08)	
Soft convexity	1 (2.0)	1 (2.0)	8 (15.7)	18 (35.3)	23 (45.1)	4.20 (0.92)	
Moderate convexity	0 (0.0)	0 (0.0)	1 (10.0)	4 (40.0)	5 (50.0)	4.40 (0.70)	
Adhesive border							0.011
No	1 (1.8)	1 (1.8)	4 (7.0)	17 (29.8)	34 (59.6)	4.44 (0.85)	
Yes	2 (4.3)	1 (2.1)	11 (23.4)	16 (34.0)	17 (36.2)	3.96 (1.04)	
Viewing option							0.437
No	1 (2.0)	1 (2.0)	10 (20.4)	15 (30.6)	22 (44.9)	4.14 (0.96)	
Yes	2 (3.6)	1 (1.8)	5 (9.1)	18 (32.7)	29 (52.7)	4.29 (0.98)	
Ostomized time							0.350
<1 week	0 (0.0)	0 (0.0)	0 (0.0)	1 (20.0)	4 (80.0)	4.80 (0.45)	
1 week–1 month	0 (0.0)	0 (0.0)	4 (25.0)	7 (43.8)	5 (31.3)	4.07 (0.77)	
1 month–3 months	0 (0.0)	1 (3.0)	7 (21.2)	10 (30.3)	15 (45.5)	4.18 (0.88)	
3 motnhs–6 months	1 (4.8)	0 (0.0)	0 (0.0)	7 (33.3)	13 (61.9)	4.48 (0.93)	
6 months–12 months	2 (6.9)	1 (3.4)	4 (13.8)	8 (27.6)	14 (48.3)	4.07 (1.20)	

## Data Availability

The datasets used and/or analyzed during the current study are available from the corresponding author on reasonable request.

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
