# Peer review of "Evaluation of Clinical Results regarding Peristomal Skin Health Associated with the Adjustment and Formulation of the New Moderma Flex One-Piece Ostomy Devices"

_jpm, 2023, doi:10.3390/jpm13020219_

Round 1

Reviewer 1 Report

Dear team, this is an organized and relevant paper. Please comment on and about costs and financial coverage of devices pre-study and post- study from the clients perspective. Though you many not have collected this in this study, financial coverage of products to patients is why readers will read your important work. I.E., who paid??? throughout and after the study.

Please publish table upright figure:1 - please do not publish side-ways as it requires costly software that some do not have to read the table.

This group of patients could be revisted (with REB) and see how they are progressing - this would be interesting and relevant.

Thank you, keep up the good work.

Author Response

RESPONSE TO REVIEWER #1

21 January 2023

Dear Editor JPM

Thank you very much for the opportunity to revise and improve the Manuscript entitled "Evaluation of clinical results regarding peristomal skin health associated with the adjustment and formulation of the new Moderma Flex one-piece ostomy devices”

            First, we want to express our sincere thanks to the reviewers, as well as their constructive comments.

            Following this letter, we detail point-by-point our comments to the suggestions of the reviewers and the changes that we have made in the revised version of our manuscript. In the revised manuscript, we have highlighted the modifications made to the original text.

We look forward to hearing from you at your earliest convenience.

Sincerely,

Julian Rodriguez Almagro

REVIEWER 1

Dear team, this is an organized and relevant paper. Please comment on and about costs and financial coverage of devices pre-study and post- study from the clients perspective. Though you many not have collected this in this study, financial coverage of products to patients is why readers will read your important work. I.E., who paid??? throughout and after the study.

Response: Dear reviewer the costs of using these devices were financed by the company Hollister. However, once the study is completed, the expenses derived from the use of devices are financed totally or partially by the public health system or private health insurance, depending on each patient. This has been included in the funding section.

Please publish table upright figure:1 - please do not publish side-ways as it requires costly software that some do not have to read the table.

Response: The suggested change has been carried out.

This group of patients could be revisted (with REB) and see how they are progressing - this would be interesting and relevant.

Response: Thank you very much for your suggestion, we will continue to work with these patients to know their evolution.

Thank you, keep up the good work.

Reviewer 2 Report

The article is well written and well exposed, and concerns the study of a registered device for the management of the different types of ostomies.

My suggestions are:

1) Authors should specify the trademark and producer name when naming for the first time the Moderna Flex device

2) Which criteria have been used to determine exclusion criteria regarding the patients' cognitive capacity? Were they exclusively subjective?

3) How are event proportions determined in the first instance to design sample size? According to which study? 

Finally, however well written, the study would be more descriptive of the subject's characteristics than a clinical study. This is because the population on which it was studied is very inhomogeneous on the reported and not reported characteristics. There is no comparison group, as already specified by the authors.

Specifically, when we talk about the degree of improvement of the skin perceived by the patient, this is evaluated on patients who have applied the different types of Moderna flex devices on different types of ostomies, which has been partially stated in the article, but above all with different initial with other potentially heterogeneous devices to manage the ostomies previously used and not described in the initial population characteristics.

If we want to say that the device has seen an improvement in the conditions of the peristomal skin, it would be appropriate to specify, if possible, when and if they were managed using other devices or if it was applied in the first instance the Moderna flex device straightforwardly after the execution of the ostomy.

Author Response

RESPONSE TO REVIEWER #2

21 January 2023

Dear Editor JPM

Thank you very much for the opportunity to revise and improve the Manuscript entitled "Evaluation of clinical results regarding peristomal skin health associated with the adjustment and formulation of the new Moderma Flex one-piece ostomy devices”

            First, we want to express our sincere thanks to the reviewers, as well as their constructive comments.

            Following this letter, we detail point-by-point our comments to the suggestions of the reviewers and the changes that we have made in the revised version of our manuscript. In the revised manuscript, we have highlighted the modifications made to the original text.

We look forward to hearing from you at your earliest convenience.

Sincerely,

Julian Rodriguez Almagro

REVIEWER 2

The article is well written and well exposed, and concerns the study of a registered device for the management of the different types of ostomies.

Response: Thank you very much for your comments.

My suggestions are:

1) Authors should specify the trademark and producer name when naming for the first time the Moderna Flex device

Response. We have made the suggested change.

2) Which criteria have been used to determine exclusion criteria regarding the patients' cognitive capacity? Were they exclusively subjective?

Response: Subjective criteria based on the assessment of the stomatherapist nurse were used.

3) How are event proportions determined in the first instance to design sample size? According to which study? 

Response: The incidence of complications in the perietomal skin of the systematic review by Malik et al "Peristomal skin complications were most common in patients with a loop ileostomy (median 14.0%)" was used as a criterion.  Complications of ileostomy were used because it is the smallest and therefore more conservative. The reduction in complications was estimated by the researchers because we considered that to be clinically relevant, the improvement should be halved. We have included the quote in that method section.

4)Finally, however well written, the study would be more descriptive of the subject's characteristics than a clinical study. This is because the population on which it was studied is very inhomogeneous on the reported and not reported characteristics. There is no comparison group, as already specified by the authors.

Specifically, when we talk about the degree of improvement of the skin perceived by the patient, this is evaluated on patients who have applied the different types of Moderna flex devices on different types of ostomies, which has been partially stated in the article, but above all with different initial with other potentially heterogeneous devices to manage the ostomies previously used and not described in the initial population characteristics.

If we want to say that the device has seen an improvement in the conditions of the peristomal skin, it would be appropriate to specify, if possible, when and if they were managed using other devices or if it was applied in the first instance the Moderna flex device straightforwardly after the execution of the ostomy.

Response: We completely agree with your comments, we will include it as a limitation. Thanks a lot.